# Native Plant Growth-Promoting Rhizobacteria Containing ACC Deaminase Promote Plant Growth and Alleviate Salinity and Heat Stress in Maize (*Zea mays* L.) Plants in Saudi Arabia

**DOI:** 10.3390/plants14071107

**Published:** 2025-04-02

**Authors:** Madeha A. Alonazi, Hend A. Alwathnani, Fahad N. I. AL-Barakah, Fahad Alotaibi

**Affiliations:** 1Plant and Microbiology Department, College of Science, King Saud University, P.O. Box 2460, Riyadh 11451, Saudi Arabia; malonze@ksu.edu.sa (M.A.A.); wathnani@ksu.edu.sa (H.A.A.); 2Department of Soil Science, College of Food and Agriculture Sciences, King Saud University, P.O. Box 2460, Riyadh 11451, Saudi Arabia; fanalotaibi@ksu.edu.sa

**Keywords:** abiotic stress tolerance, ACC deaminase, *Bacillus*, bacterial inoculants, maize, plant growth promotion

## Abstract

Halotolerant, plant growth-promoting rhizobacteria (PGPR) are known to alleviate plant growth under abiotic stresses, especially those isolated from saline arid soils. In this study, 66 bacterial isolates, obtained from various habitats in Saudi Arabia, were characterized for their plant growth-promoting (PGP) traits, and screened for heat and salt stress resilience. Finally, selected halotolerant PGPR strains were assessed for their potential to improve maize (*Zea mays* L.) growth under salinity stress using in vitro assays. Our results indicated that many isolates possessed key PGP traits such ACC deaminase, N-fixation, and phytohormone production. Additionally, several isolates were able to tolerate high temperatures, and 20 bacterial isolates were classified as halotolerant. Furthermore, among the isolates, *Pseudomonas soyae* (R600), *Bacillus haynesii* (SFO145), *Salinicola halophilus* (SFO075), and *Staphylococcus petrasii* (SFO132) significantly enhanced various maize growth parameters under salt stress conditions when compared to uninoculated plants. These halotolerant PGPR are good candidates to be explored as bioinoculants for sustainable agriculture under saline arid soil conditions.

## 1. Introduction

Among all grains, maize (*Zea mays* L.) has the largest planting area worldwide, with an average of 200 million hectares [1], and is one of the most produced and consumed crops in the world [2]. It has high economic/social value for many communities; in numerous countries, maize also is a primary consumed food [2]. Additionally, maize is a main ingredient in a variety of industrial products including dextrose, maize starch, and maize syrup [3]. Maize is sensitive to salinity [4]. Like most crops, salinity negatively affects a maize plant’s relative growth rate, osmotic status, transpiration, ion transport, photosynthetic activity, and senescence [5,6]. In addition, the excessive accumulation of salt in a maize plant’s cells can produce reactive oxygen species (ROS), such as hydrogen peroxide, hydroxyl radicals, and superoxide anions, all of which inhibit photosynthetic activity [7,8].

Salinity is a major detrimental stressor that hinders plant production and consequently affects the ability to meet the increasing food demand of the world’s fast-growing population. It has been estimated that almost 20% of cultivated lands are affected globally by hyper-salinity, causing crops to suffer oxidative stress, ion toxicity, and osmotic stress [9]. For example, it has been estimated that salinity stress decreased the yield of major crops such as maize, rice, wheat, and barley by up to 70% [10]. As a matter of fact, there is an urgent need to increase agricultural productivity per unit area. Naturally occurring bacteria that are located in the soil near plant roots (e.g., the rhizosphere) have shown tremendous potential to alleviate salinity stress [9]. These bacteria are called plant growth-promoting rhizobacteria (PGPR). PGPR are free-living, soil-borne bacteria that are isolated from the rhizosphere and represent a wide range of diverse phyla [11,12]. PGPR have been extensively documented to positively affect plants through a variety of metabolic activities through which they can have a direct or indirect impact on host plant growth and development [13,14].

Indirectly, PGPR minimize or even eliminate the harmful effects of plant pathogens by producing antimicrobial compounds or by enhancing the host’s natural resistance [15]. The direct mechanisms of plant growth promotion include the following: increasing nutrient acquisition such nitrogen fixation, phosphorus solubilization, iron sequestration through siderophore production; altering the endogenous concentration of substances known to regulate plant growth (PGRs), such as auxin, gibberellins, cytokinins, ethylene, and abscisic acid [16,17]; and alleviation of tolerance to various biotic and abiotic stresses through the production of the bacterial enzyme aminocyclopropane-1-carboxylic acid (ACC) deaminase [18].

Both biotic and abiotic stresses trigger higher ethylene production, with salinity being the most significant of these variables. Excessive ethylene production, caused by various stresses, can hinder root development and ultimately reduce plant growth and yield [19]. There exist multiple strategies to mitigate the elevated levels of ethylene produced during stressful situations, which may involve chemical or biological approaches [20]. In chemical procedures, a certain chemical act as an inhibitor of ethylene activity or of ethylene synthesis. Although these compounds have been proven to be effective in reducing ethylene levels in plants, they are often costly, have limited usefulness, or cause potential harm to the environment [21]. A biological approach is another tactic, including the utilization of PGPR as plant inoculant, which contain ACC deaminase to promote plant growth under stressful conditions [19,22,23].

ACC deaminase is an enzyme commonly found in many PGPR isolates. The enzyme ACC deaminase is capable of degrading the ethylene precursor, ACC, to ammonia and α-ketobutyrate, which decreases the amount of ethylene in plants [14]. The stress-induced ethylene-mediated negative effects on plants can be reduced or completely eliminated by bacterial strains expressing the ACC deaminase phenotype [24]. Moreover, using ACC deaminase-producing PGPR in soil and plant systems is more practical, affordable, and environmentally benign than other methods. Furthermore, the ACC deaminase phenotype is found in numerous PGPR species that naturally inhabit the rhizosphere. This allows them to thrive in a variety of conditions within the rhizosphere and rhizoplane, making the utilization of PGPR with ACC deaminase activity highly advantageous [25]. In addition, by manipulating the regulation of ethylene production in response to various environmental factors such as salt, high temperature, drought, water logging, and pollutants, ACC deaminase-containing PGPR can help enhance plant growth, particularly in challenging conditions [12,26].

Utilizing ACC deaminase-producing PGPR could prove to be a valuable strategy for improving the growth and production of maize when faced with salt stress conditions [27]. Therefore, in addition to traditional breeding and genetic engineering, the incorporation of PGPR could be advantageous in developing strategies to improve maize growth under conditions of salinity [28,29].

The selective pressure derived from the misuse of chemical fertilizers, pesticides, and genetic breeding of varieties of plants may have resulted in the loss of the diversity of beneficial PGPR associated with the rhizosphere of domesticated plants grown under intensive agriculture [30]. Therefore, we hypothesized that native bacterial PGPR isolates with ACC deaminase activity obtained from the rhizosphere of native plants adapted to harsh environments of Saudi Arabia would alleviate salt stress impact and enhance maize growth. Our aims in this study were to assess the potential of different bacterial isolates from various habitats in Saudi Arabia to produce ACC deaminase, to characterize bacterial isolates for various PGP traits, and to evaluate the capacity of bacterial isolates to withstand salt and heat stresses. Additionally, selected bacterial strains were further evaluated regarding their ability to improve the germination and growth of maize seeds under normal and saline conditions. Halotolerant bacterial isolates with multiple PGP activities could be further utilized in biotechnological applications to enhance agricultural productivity in saline soils.

## 2. Results

### 2.1. Plant Growth-Promoting Characteristics of Bacterial Isolates

Most isolates produced multiple plant growth-promoting characteristics; among the 66 isolates tested, 54 were capable of producing NH_3_ (81.8%), 54 were capable of fixing N (81.8%), ACC deaminase was produced by 50 bacterial isolates (75.75%), 38 isolates synthesized IAA (57.58%), and 20 bacterial isolates solubilized P (30.3%) (Figure 1). Interestingly, five bacterial isolates (7.5%) possessed all the PGP traits under investigation: *Salinicola halophilus* (SFO075), *Bacillus haynesii* (SFO145), *B. spizienii* (Z180), *B. stercoris* (K2), and *Pseudomonas thivervalensis* (K26) (Figure 1).

### 2.2. Characterization of Bacterial Isolates for Heat and Salt Tolerance

Among the 66 isolates tested in this study, 37 were able to tolerate heat stress (56%), as shown in Appendix A. Interestingly, four bacterial isolates shared all PGP traits and tolerated heat stress (Appendix A)—namely, *B. haynesii* (SFO145), *B. spizienii* (Z180), *B. stercoris* (K2), and *P. thivervalensis* (K26). Similarly, the majority of isolates could grow at 2.5% NaCl and 5% NaCl with varying growth patterns (Appendix A). Interestingly, based on their growth pattern at different NaCl concentrations, 20 bacterial isolates were classified as highly halotolerant (Appendix A). Interestingly, three bacterial isolates shared all five PGP traits and tolerated salinity—namely, *S. halophilus* (SFO075), *B. haynesii* (SFO145), and *B. spizienii* (Z180) (Appendix A). Also, our results revealed that *B. haynesii* (SFO145) and *B. spizienii* (Z180), which both grew at 50 °C and 10% NaCl, were able to display all five PGP traits (Appendix A) and were considered as multi-stress-tolerant strains.

### 2.3. Seed Germination Test

Based on their PGP characteristics, and tolerance to salinity and heat, a subset of bacterial isolates (*n* = 30) was selected for the seed germination test (Table 1). In total, 30% of bacterial isolates (*n* = 10) significantly enhanced shoot length compared to the uninoculated control treatment (Appendix A). Similarly, several bacterial strains significantly increased the root length compared to the uninoculated control treatment (Appendix A). Additionally, nine bacterial isolates significantly increased seed germination (%) compared to the uninoculated control treatment (Appendix A), while several bacterial isolates showed a statistically significant high vigor index value when compared to the uninoculated control treatment (Figure 2). Notably, bacterial strains *B. haynesii* (SFO145), *S. halophilus* (SFO075), *S. pasteuri* (SFO132), and *P. soyae* (R600) significantly enhanced all the above-mentioned seed germination parameters compared to the uninoculated control treatment (Figure 2 and Appendix A).

### 2.4. Screening Assay of Isolates to Enhance Salt Stress Tolerance

The results of the control seed germination of five selected isolates, *B. haynesii* (SFO145), *P. soyae* (R600), *S. pasteuri* (SFO132), *S. halophiles* (SFO075), and *B. inaquosorum* (SFO119), are provided in Figure 3, Figure 4, Figure 5 and Figure 6.

Under non-saline conditions (0 mM NaCl), two bacterial treatments *S. halophiles* (SFO075) and *S. petrasii* (SFO132) significantly increased the shoot length by up to 69.31% and 69.31% as compared to the respective uninoculated control (Figure 3), while inoculation with *B. inaquosorum* (SFO119) significantly decreased shoot length up to 92.06% as compared to the uninoculated control treatment (Figure 3). Under saline conditions (100 mM NaCl), inoculation with *B. inaquosorum* (SFO119) also significantly decreased the shoot length by up to 97.87%, while seed inoculation with *B. haynesii* (SFO145), *S. petrasii* SFO132, and *S. halophilus* (SFO075) significantly increased the shoot length by up to 576.6%, 242.55%, and 227.66% as compared to the uninoculated control treatment, respectively (Figure 3).

Under non-saline conditions (0 mM NaCl), only *B. inaquosorum* (SFO119) treatment significantly decreased root length by up to 71.07% as compared to uninoculated control treatment (Figure 4), while inoculation with *P. soyae* (R600), *S. petrasii* (SFO132), *S. halophiles* (SF075), and *B. haynesii* (SFO145) significantly increased the root length by 245.91%, 227.04%, 214.47%, and 189.31% as compared to the uninoculated control treatment, respectively (Figure 4). Under saline conditions (100 mM NaCl), the inoculation with *B. inaquosorum* (SFO119) also significantly decreased the root length by up to 98.92% as compared to the respective uninoculated control treatment (Figure 4), while seed inoculation with *B. haynesii* (SFO145) significantly increased the root length by up to 300% as compared to the uninoculated control treatment (Figure 4). Under non-saline conditions (0 mM NaCl), inoculation with *B. inaquosorum* (SFO119) significantly decreased seed germination (%) by up to 44% compared to the uninoculated control treatment (Figure 5), while the rest of the inoculation treatments showed no change in seed germination (%) as compared to the uninoculated control (Figure 5). Under saline conditions (100 mM NaCl), inoculation with *B. inaquosorum* (SFO119) also significantly decreased seed germination (%) by up to 86.36% compared to the uninoculated control treatment (Figure 5), while seed inoculation with *B. haynesii* (SFO145), *P. soyae* (R600), and *S. halophiles* (SFO075) increased seed germination (%) by up to 13.64% as compared to the uninoculated control treatment (Figure 5).

Under non-saline conditions (0 mM NaCl), only inoculation with *B. inaquosorum* (SFO119) significantly decreased the vigor index by up to 88.79% as compared to the uninoculated control treatment (Figure 6), while inoculation with *S. petrasii* (SFO132), *S. halophiles* (SFO075), *P. soyae* (R600), and *B. haynesii* (SFO145) significantly increased the vigor index by up to 141.38%, 135.63%, 92.96%, and 89.37% as compared to the uninoculated control treatment, respectively (Figure 6). Under saline conditions (100 mM NaCl), inoculation with *B. inaquosorum* (SFO119) also decreased the vigor index by up to 99.67%, while seed inoculation with *B. haynesii* (SFO145), *S. halophiles* (SFO075), and *S. petrasii* (SFO132) significantly increased the vigor index by up to 462.81%, 215.82%, 192.01% as compared to the uninoculated control treatment, respectively (Figure 6). Figure 7 shows the seed germination of maize seeds in non-salt (½ MS) and in salt (½ MS + 100 mM) conditions.

### 2.5. Molecular Characterization

Bacterial isolates were identified to the genus or species level by phylogenetic and similitude analyses. Though there is no consensus on the threshold for differentiating bacterial species, in this study a 16S rRNA sequence similarity above ≥98.65% was used to define the species [31]. Bacterial isolates were classified into 25 genera, belonging to the phyla Baclliota, Pseudomonasdota, Actionmycota, and Bacteroidota via Sanger sequencing of the 16S rRNA gene The most abundant genus was *Bacillus* (25.76%), followed by *Pseudomonas* (15.15%), *Staphylococcus* (15.15%), *Paenibacillus* (4.54%), *Stenotrophomonas* (4.54%), *Arthrobacter* (3.03%), *Chryseobacterium* (3.03%), *Paenarthrobacter* (3.03%), *Acinetobacter* (1.52%), *Achromobacter* (1.52%), *Cellulomonas* (1.52%), *Cytobacillus* (1.52%), *Enterobacter* (1.52%), *Ensifer* (1.52%), *Klebsiella* (1.52%), *Kocuria* (1.52%), *Metabacillus* (1.52%), *Microbacterium* (1.52%), *Nocardioides* (1.52%), *Priestia* (1.52%), *Psychrobacter* (1.52%), *Stutzerimonas* (1.52%), *Salinicola* (1.52%), *Sphingobacterium* (1.52%), and *Streptomyces* (1.52%).

## 3. Discussion

This study explored the PGP potential of native bacterial isolates obtained from various soils in Saudi Arabia. Sixty-six bacterial isolates from contrasting, different soil environments (i.e., mangrove rhizosphere, olive rhizosphere, and Acacia rhizosphere) were screened for their PGP activities as well as for their ability to tolerate salt and heat stresses. The most effective PGP isolates were selected and further tested to ameliorate the negative impact of salt stress on the growth of maize plants. The soil of Saudi Arabia, similar to that of other arid regions, is characterized as sandy with extremely low organic matter, having very low nutrients and high salt content [32,33]. Therefore, it is necessary to search for potential PGP bacterial inoculants that can improve plant growth under adverse abiotic stresses such salinity, drought, and high temperatures.

Phylogenetic analysis of 66 rhizobacterial isolates using 16S rRNA partial sequences revealed that the identified isolates were highly diverse at the genus level. The majority of the strains belonged to three different genera including *Bacillus*, *Staphylococcus*, and *Pseudomonas*. These genera are commonly isolated from the rhizosphere of plants in arid and semi-arid soils [34,35,36]; they are well known as halotolerant PGPR [10,36,37,38] and our observations are in line with these earlier reports. PGPR play a vital role in stimulating plant growth and development in several ways [39,40,41,42]. In the present study, 66 isolates were assessed for the following PGP traits, including ACCD activity, IAA production, P-solubilization, N-fixation, and NH_3_ production.

IAA is an important phytohormone with the capacity to promote cell differentiation, seedling growth, and the elongation and formation of lateral roots in plants [43,44,45]. Our results revealed that 57% of the tested bacterial isolates produced IAA. In the present study, ammonia production and N-fixation were the most common PGP traits among the culture collection, with 82% of isolates testing positive for both phenotypes. NH3 production and N-fixation are essential traits linked to PGP through supplying N to their host plants, thereby promoting root and shoot elongation and subsequent biomass production [46,47]. Similar results in many studies have proven that PGPR have been used mostly to enhance plant growth through certain mechanisms such as the production of plant growth regulators and the fixation of nitrogen [48].

Phosphorus is an important macronutrient for crop growth and development. Phosphorus is immobilized and generally remains unavailable to plants due to fixation with Ca^+2^ in calcareous soils [49,50]. Thus, applying phosphate solubilizing microorganisms is a cost-effective alternative for making this mineral available to plants [42,51]. Our results revealed that 30% of strains have the ability to solubilize P and therefore, could be effective in improving crop yield.

The mechanism of action of ACC deaminase-producing bacteria in the improvement of both biotic and abiotic stresses involves a reduction in ethylene levels through the activity of the enzyme ACC deaminase which breaks down ACC into α-ketobutyrate and ammonia instead of ethylene [52]. Bacteria producing ACC deaminase activity are known to improve the growth of a wide range of plants under stressful conditions like drought, salinity, heavy metals, and flooding [45,53,54,55]. In our study, 75% of isolates were able to produce ACCD. Halotolerant PGPR obtained from saline environments appear to maintain ACCD activity which helps plants overcome salinity stress via a reduction in ethylene levels [56]. For example, 25 out of 140 halotolerant PGPR from coastal soils of the Yellow Sea showed ACCD activity [57].

Several isolates (50%) in our study showed high tolerance toward heat stress. High temperatures have been shown to negatively affect crop yield, seed germination, seedling development, and plant metabolism [58,59]. In our study, thermotolerant bacterial isolates were mainly affiliated with the Bacillota phyla, represented by *Bacillus* and *Staphylococcus* (Appendix A). Similar results were reported previously regarding the ability of these genera to tolerate high-temperature stress, largely attributed to their ability to form resistant endospores, which allow them to survive harsh conditions [60]. However, heat tolerance in these bacterial genera is likely driven by a combination of mechanisms beyond spore formation. While spore formation plays a significant role in the survival of *Bacillus* and related genera, other mechanisms such as the production of heat-shock proteins (HSPs), membrane adaptations, antioxidant systems, and DNA repair processes also contribute to their thermotolerance. These adaptive responses may vary across genera, highlighting the complexity of heat tolerance mechanisms in bacteria. The harmful impacts of severe temperatures on crops can be mitigated by the use of thermotolerant plant growth-promoting bacteria [61]. For example, Khan et al. [62] reported that *Bacillus cereus SA1* enhances the growth of *soybean* under heat stress; similarly, *B. cereus* was reported to minimize the impact of heat stress on *Solanum lycopersicum* development.

Many of the bacterial isolates in this study (45%) exhibited high tolerance to different levels of NaCl. To proliferate and endure in salty environments, halotolerant bacteria use a variety of techniques, such as limiting the uptake of salt due to the composition of the cell wall; controlling intracellular ion concentrations by pumping ions out of the cell through K+/Na+ ion transporters; producing proteins and enzymes that are adapted to high concentrations of solute ions; and producing exopolysaccharides (EPSs) that aid in the development of hydrating biofilms [63,64,65]. In our study, halotolerant bacterial isolates which grow at the 10% NaCl concentration were mainly affiliated with the genus *Bacillus* (Appendix A). Other genera with high NaCl tolerance include *Kocuria*, *Salinicola*, *Metabacillus*, *Staphylococcus*, and *Arthrobacter* (Appendix A). Our results suggest that these bacteria are halophiles growing best in media containing high levels of NaCl [38,66]. The halotolerant characteristics of these bacteria make them potential candidates for growing crops in saline soils or with saline irrigation.

Interestingly, several isolates in our study were able to exhibit multiple PGP activities in addition to their tolerance to heat and salt stresses (Figure 1 and Appendix A). For example, *Bacillus spizienii* (Z180), *B. haynesii* (SFO145), *B. stercoris* (K2), and *Pseudomonas thivervalensis* (K26) were found to be thermotolerant PGPR (Appendix A). Similarly, *Salinicola halophilus* (SFO075), *B. spizienii* (Z180), and *B. haynesii* (SFO145) were halotolerant PGPR (Appendix A). Surprisingly, two isolates were able to tolerate heat and salt, and possessed all PGP traits; specifically, *B. haynesii* (SFO145) and *B. spizienii* (Z180) (Appendix A). In agreement with our results, Bokhari et al. [67] reported the ability of various PGP *Bacillus* strains to remain resilient to both heat and salinity. Similarly, Upadhyay et al. [68] revealed that the majority of halotolerant PGP bacterial isolates from the rhizosphere of bread wheat growing under saline conditions belonged to the class Bacilli. The PGP traits, together with the resilience to both heat and salinity, possessed by the bacterial isolates reported in our study make them good candidates for sustainable agriculture [62,67,69,70].

As an initial screening for the PGP potential, we tested the effects of 30 selected bacterial isolates on maize seed gemination variables under in vitro conditions. Our results indicated that several bacterial strains, when compared with the uninoculated control treatment, significantly increased seed germination, shoot length, root length, and vigor index parameters, notably, strains *P. soyae* (R600), *B. haynesii* (SFO145), *S. halophilus* (SFO075), *Arthrobacter cheniae* (B1105), *Staphylococcus petrasii* (SFO132), *S. pasteuri* (SFO112), and *B. stercoris* (K2). In agreement with our results, previous reports indicated that the germination of the seeds of various crops were either inhibited, stimulated, or remained non-affected upon inoculation with bacterial inoculants [71,72,73]. Similar improvement of seed germination variables by PGPR has been reported in maize [74,75] and pearl millet [76]. These findings may be best explained on the grounds of their multi-PGP traits activity, such as the synthesis of endogenous plant growth regulators (e.g., auxins and cytokinins) which can promote cell division and elongation [72,77]. It should be noted, however, that some isolates inhibited various seed germination variables in our study. This can be attributed to several reasons, including the overproduction of high levels of plant phytohormones, secretion of inhibitory molecules, or production of phenazine by an inoculant [61,73,78]. For example, it has been reported that *Azospirillum* sp. secrete inhibitory molecules, which block the germination of striga seeds [79]. Similarly, the inhibition of seed germination by *Pseudomonas* spp. has been attributed to the production of phenazine [73].

Finally, after several rounds of screening, we tested the effects of five selected bacterial isolates on maize seed gemination variables under saline and non-saline conditions under in vitro settings. Our results revealed that *S. halophilus* (SFO075), *S. petrasii* (SFO132), *B. haynesii* (SFO145), and *P. soyae* (R600) significantly enhanced various maize variables under both saline and non-saline stress when compared to the uninoculated control treatment (Figure 3, Figure 4, Figure 5 and Figure 6). Thus, the plant growth-promoting effects observed in the current work probably rely more on the possession of multiple PGP traits, including ACC deaminase [80]. Interestingly, *B. haynesii* (SFO145), originally isolated from a mangrove environment [35] with a naturally high salinity level [81], demonstrated a remarkable ability to increase the shoot length of maize under saline conditions, even more so than under optimal conditions. This observation aligns with previous reports suggesting that certain PGPR strains exhibit enhanced effectiveness under abiotic stress conditions, likely due to their capacity to activate specific beneficial mechanisms in response to environmental stressors such as salinity [82]. In summary, the PGPR activity of *B. haynesii* (SFO145) is likely linked to its genetic and metabolic adaptations to high-salinity environments. Its full potential as a growth promoter is realized under salt stress, where it can activate beneficial mechanisms that improve plant growth, whereas under non-saline conditions, these mechanisms may not be induced, limiting its effectiveness as a PGPR. de Zélicourtet et al. [83] identified that the ability of *Enterobacter* sp. SA187 to alleviate salt stress in *A. thaliana* was due to its ability to produce 2-keto-4-methylthiobutyric acid (KMBA) which stimulates the ethylene signaling pathway in Arabidopsis. Further studies at the genome, metabolome, and transcriptome levels are crucial to decipher the mode of action of the isolate *B. haynesii* (SFO145). On the other hand, *B. inaquosorum* (SFO119) did not increase the growth of maize under both saline and non-saline conditions despite their possession of multiple PGP activities. This finding indicates that bacterial isolates displaying PGP activities under in vitro conditions may not necessarily function as PGPR with host plants in natural soil conditions, as their effectiveness can be influenced by complex soil interactions, plant–microbe specificity, and environmental factors [84,85].

In this study, the possession of multi-PGP traits along with the capacity of *B. haynesii* (SFO145), *P. soyae* (R600), *S. halophilus* (SFO075), and *S. petrasii* (SFO132) to enhance plant tolerance toward salinity stress makes them ideal candidates for sustainable farming. However, to reach this ultimate goal, extensive evaluation of the potential of these bacterial isolates with a variety of crops under abiotic stresses will be essential, including field trials on different saline soils and saline irrigation. Additionally, the inconsistent reproducibility of PGPR inoculants under real field conditions is a major obstacle in developing biofertilizers based on a single strain, as these inoculants fail to compete with indigenous soil microbiomes due to soil type and other environmental and climatic factors. Recent approaches, such as the development of synthetic microbial communities (SynCom) that interact synergistically with the native microbiomes already present in the soil, could overcome the limitations associated with individual bioinoculants [86].

## 4. Materials and Methods

### 4.1. Source of Bacterial Isolates

Sixty-six bacterial isolates (from the culture collection of the Soil Microbiology Laboratory, Soil Science Department, College of Food and Agriculture Sciences, King Saud University) were obtained for this study. The bacterial isolates used in this study represent diverse habitats such as the rhizospheres of wild *Acacia* sp., mangrove (*Avicennia marina*), and olive trees (consult Appendix A for more information regarding the origin and the soil characteristics from which the bacteria were isolated). The bacterial isolates were revived from the stock cultures by cultivating them in 50 mL half-strength Trypticase Soy Broth (TSB) at room temperature for 48 h. The culture was constantly agitated at 150 rpm on a rotary shaker.

### 4.2. Screening for Plant Growth-Promoting Activities

#### 4.2.1. ACCD Activity

The method described previously by Penrose and Glick (2003) was employed to estimate the ACCD activity of the isolates. Briefly, 100 µL of an overnight culture of each isolate was transferred to tubes containing 5 mL of half-strength TSB medium and incubated for 48 h at 28 °C in a rotary shaker, using constant agitation at 150 rpm. A loopful of each bacterial isolate growing in liquid culture was streaked on a DF minimal-salt agar plate containing 3 mM ACC solution, which was spread onto the agar plate immediately prior to use. Plates were incubated at 28 °C for up to 1 week. The presence of growth in the DF-ACC agar plates was considered positive [87,88]. The bacterial strains were ranked according to four categories: − growth, no growth; + growth, mild growth; ++ growth, significant growth; and +++ growth, rapid growth. The experiment was conducted three times, each in duplicate. The above-mentioned parameters were set for subsequent experiments.

#### 4.2.2. Evaluation of Phosphate Solubility and Screening

The bacterial isolates were tested for their ability to solubilize inorganic phosphate using qualitative assays, following the methods described in Nautiyal [89]. The isolates were cultured in ½ TSB medium and incubated in a rotary shaker (model: SI9R-2, Shel Lab, Sheldon Manufacturing, Inc., Cornelius, OR, USA) at 120 rpm at 28 °C for 48 h. Subsequently, a small portion of the bacterial culture was precisely placed in the center of an agar plate (National Botanical Research Institute’s phosphate growth medium (NBRIP)). The halo and colony diameters were observed after 14 days of incubation of the plates at 28 °C. The appearance of clear zones surrounding colonies indicated phosphate solubilization.

#### 4.2.3. Indole 3-Acetic Acid (IAA) Production

The IAA activity of bacterial strains was estimated according to Ribeiro and Cardoso [90]. The isolates were cultivated in 5 mL of DF minimum-salt medium supplemented with tryptophan (1 mg·mL^−1^) as an auxin precursor. Later, the tubes were incubated in a rotary shaker at 120 rpm for 2 d at 28 °C. After incubation, the bacterial cultures were centrifuged at 9500× *g* for 10 min. Salkowski’s reagent (100 μL) was added to the tubes with the supernatant, and the sample was kept undisturbed at ambient temperature in a dark environment for 30 min. The formation of a pink color indicated the production of IAA. To quantify the IAA, absorbance measurements were taken at 535 nm using a UV/Visible spectrophotometer (Genesys 20, Thermo Scientific, Waltham, MA, USA). The IAA concentration was estimated with a standard curve ranging from 0 to 0.5 mg mL^−1^ of IAA.

#### 4.2.4. Nitrogen Fixation

The ability of bacterial isolates to proliferate on a combined carbon medium (CCM) lacking N was assessed [91]. The bacterial isolates were cultivated in ½ TSB in a rotary shaker at 28 °C for 48 h while being shaken continuously at 150 rpm. Each bacterial isolate grown in liquid culture was spread onto a N-deficient CCM agar plate, which was subsequently incubated at 28 ± 0.1 °C for a maximum of one week. The capacity of the agar plates to promote growth was found to be favorable.

#### 4.2.5. Production of Ammonia

The process described by Cappuccino and Sherman [92] was used to assess the ammonia generated by the isolates. The isolates were initially inoculated and cultivated for 48 h at 28 °C in tubes containing 10 mL of peptone water. Following the incubation period, 1.5 mL microtubes were filled with 1 mL of each culture. Then, 50 μL of Nessler’s reagent was added to each microtube, and changes in color were carefully observed to confirm the possible presence of ammonia. A light-yellow hue suggests a moderate amount of ammonia, while a deeper yellow to brownish shade suggests the potential for ammonia synthesis.

#### 4.2.6. Effect of Temperature on PGP Traits

The bacterial isolates were tested to evaluate how well they tolerated heat stress at various temperatures (37 °C, 45 °C, and 50 °C). The isolates were streaked onto 1/10 TSA plates and incubated for 24, 48, and 72 h at different temperatures. After the incubation phase, the growth was documented [93].

#### 4.2.7. Salinity Tolerance

The tolerance of bacterial isolates to different NaCl concentrations (0%, 2.5%, 5%, and 10%) was investigated. After streaking pure isolates onto 1/10 TSA plates supplemented with 2.5%, 5%, and 10% NaCl, the mixture was cultured for 7 d at 28 °C. Growth was observed following the incubation period [94].

### 4.3. Maize Seed Germination Assay

Following multiple rounds of screening, a total of 30 bacterial isolates were chosen for the maize seed germination assay to assess their impact on various seed parameters.

#### 4.3.1. Preparation of PGP Inoculum

For each isolate, a growth culture was initiated on 1/10 TSA plates and incubated at 28 °C for 72 h. Individual colonies of each isolate were transferred to a 50 mL Falcon tube with sterilized half-strength TSB and incubated on a rotary shaker (120 rpm) at 28 °C for 48 h. Using freshly prepared sterile phosphate-buffered saline (PBS), the optical density (OD) of the bacterial suspensions was carefully adjusted to OD = 1. The bacterial suspensions were spun in a centrifuge for 15 min at 5000× *g* to increase their concentration. Afterward, the samples were rinsed three times with PBS and then resuspended in sterile tap water. For the determination of the concentration of inoculum bacteria, the resulting cell suspension was serially diluted and spread onto 1/10 TSA plates. The plates were then incubated for 72 h at 28 °C, allowing for proper growth and observation. Approximately 10^8^ cfu mL^−1^ of inoculum was produced by the process.

#### 4.3.2. Seed Inoculation with Bacterial Isolates

Maize seeds (cv. Hybrid 168) were kindly provided by the Department of Plant Production, King Saud University. Seeds were surface-disinfected by soaking in ethanol (65%) for 3 min and sodium hypochlorite (1.2%) for 5 min on a rotary shaker, followed by 10 rinses in autoclaved tap water to remove excesses ethanol and bleach [95]. To ensure complete colonization during seed germination, 40 surface-disinfected seeds were added to 30 mL of bacterial suspension and shaken at 120 rpm for 4 h at 28 °C. Control seeds were mixed with 50 mL of autoclaved distilled water.

#### 4.3.3. Seed Germination Assay

By placing the coated seeds onto sterile filter paper saturated with 4 mL of autoclaved distilled water in Petri dishes and incubating them at room temperature, we were able to examine the impact of the bacterial isolates on seed germination. Four replicate Petri dishes containing 10 inoculated seeds were used for each bacterial treatment as well as for the control treatment. The control treatment consisted of non-inoculated seeds. After 7 days post-inoculation, root length, shoot length, germination (%), and vigor index parameters were measured. Germination (%) was estimated as the percentage of germinated seeds after 40 h (day 2) and 184 h (day 8) of seed inoculation. For a seed to be considered germinated, it is necessary for the radicle to measure at least 1 mm in length. Germination (%) and the vigor index were calculated using the following formula [96,97,98]:Germination rate = n/N × 100 (%)
where (n) is the number of germinated seeds after 7 days and (N) is the total number of seeds. Along with germination percentage, root length and hypocotyl length were also recorded to calculate the vigor index using the following formula [99]:Vigor index = (mean root length + mean shoot length) × % germination.

### 4.4. Screening Selected Isolates for Their Ability to Enhance Maize Salt Stress Tolerance Under In Vitro Conditions

For the seed germination assay, the same prior methodology was used; the seeds were inoculated with bacterial isolates that exhibited statistically significant effects on maize seed germination variables from the above-mentioned step. Five isolates were selected for this assay, namely SFO145, R600, SFO132, SFO075, and SFO119, as well as the uninoculated control treatment. The inoculated seeds were sown on Petri dishes containing either half-strength (1/2) Murashige and Skoog (MS) medium (without NaCL) and ½ MS + 100 mM NaCl with 5 seeds on each plate. Five replicate plates were used for each bacterial treatment as well as for the uninoculated control treatment [83]. By the end of the 7th day, germination (5) was calculated as described above. The seedling vigor index value, seedling shoot length, and root length were recorded.

### 4.5. Identification of Bacterial Isolates

The phylogenetic identification of isolates was performed by amplifying the bacterial 16S rRNA gene via PCR using the primer pair 27F (5′AGA GTT TGA TCM TGG CTC AG-3′) [100] and 1492R (5′ TAC GGY TAC CTT GTT ACG ACT T 3′), as described previously [101]. Amplification was performed under the following PCR conditions (PCR System 2720, Applied Biosystems, Singapore): initial denaturation at 94 °C for 5 min, followed by 34 cycles of denaturation at 94 °C for 1 min, annealing at 52 °C for 1.5 min, extension at 72 °C for 2 min, and a final extension at 72 °C for 7 min. PCR products were visualized on GelRed-stained 1.5% agarose gels. The DNA Sanger sequencing service was carried out by Macrogen Inc. (Seoul, Republic of Korea). The obtained DNA sequences were trimmed, and a pair of forward and reverse reads of the 16S rRNA target was assembled using Geneious Pro V.12 (Biomatters Inc., San Diego, CA, USA). The sequences were compared with reference sequences deposited in the GenBank database using the BLAST algorithm (https://blast.ncbi.nlm.nih.gov/Blast.cgi, accessed on 20 March 2025). The partial 16S rRNA gene sequences obtained from the bacterial isolates have been deposited in GenBank under the accession numbers (PQ454066-PQ454079 and PQ461934-PQ461980).

### 4.6. Statistical Analyses

The shared features of the characterized strains were visualized using a Venn diagram, as proposed by Bardou et al. [102], to display a list comparison. Using the statistical analysis program SPSS, version 28 (2023), a one-way analysis of variance (ANOVA) was conducted to analyze the collected data for each treatment and parameter of this investigation. The generated data were reported as arithmetic mean ± SE for each treatment. Significantly different means were separated using the Duncan’s new multiple range test (DMRT) at a 5% level of significance, as adopted by Chukwuneme et al. [103].

## 5. Conclusions

In conclusion, the characterization of several ACC deaminase-producing PGPR isolated from various habitats of Saudi Arabia revealed several growth-promoting capabilities and alleviation of plant growth under saline conditions. Due to the combination of PGP capabilities together with the ability of *P. soyae* (R600), *B. haynesii* (SFO145), *S. halophilus* (SFO075), and *S. petrasii* (SFO132) to enhance maize tolerance to salt, these isolates are promising candidates for the development of next-generation bioinoculants tailored for sustainable crop production, especially in saline soils. However, rigorous greenhouse and field trials must be conducted to test their efficiency under natural soil conditions. Also, bacterial isolates characterized in this study could be an excellent starting point to build up or engineer microbial consortia with stable, robust, and predictable behaviors. Additionally, genomic, transcriptomic, and metabolomic studies of the most promising halotolerant PGPR from this study would shed light on the PGP mechanisms connected to induced salinity tolerance in maize.

## Figures and Tables

**Figure 1 plants-14-01107-f001:**
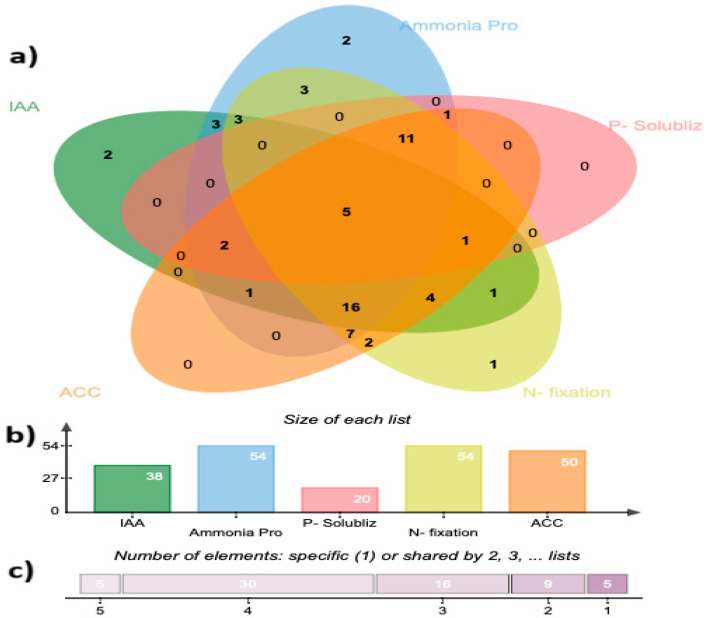
Bacterial isolates with multiple plant growth-promoting (PGP) activities. (**a**) Venn diagram illustrating the number of bacterial isolates that have one or more of the tested PGP activities, (**b**) bar graph showing the absolute numbers of isolates possessing each of the PGP traits, and (**c**) the number of isolates specific to sharing each of the five PGP traits.

**Figure 2 plants-14-01107-f002:**
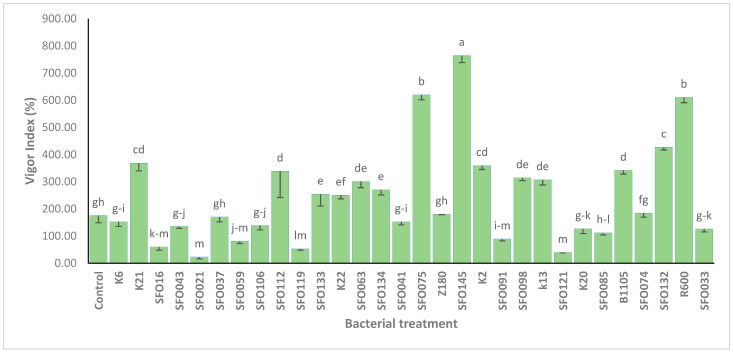
Effect of bacterial inoculants on vigor index (%) of geminated seeds of maize measured after 8 days of seed inoculation. Error bars indicate vigor index (%) mean ± SE. Lettering indicates significance values according to LSD test (*p* ≤ 0.5).

**Figure 3 plants-14-01107-f003:**
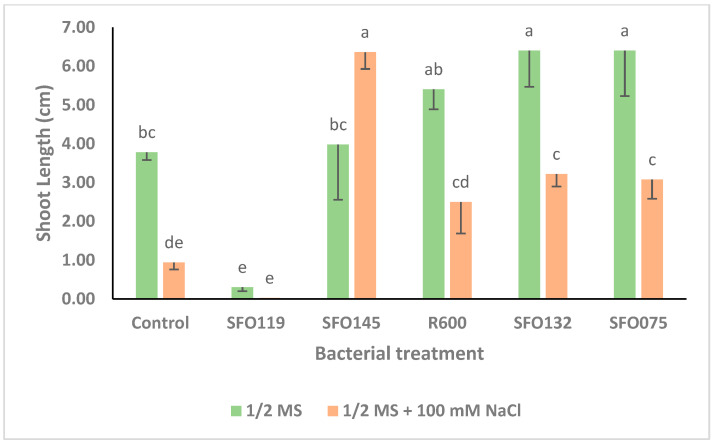
The effect of selected PGPR isolates on the shoot length (cm) of maize seeds under saline conditions (1/2 MS + 100 mM NaCl) and non-saline (1/2 MS) conditions. Error bars indicate the shoot length (cm) mean ± SE. The lettering indicates the significance status among the means by the LSD test (*p* ≤ 0.5).

**Figure 4 plants-14-01107-f004:**
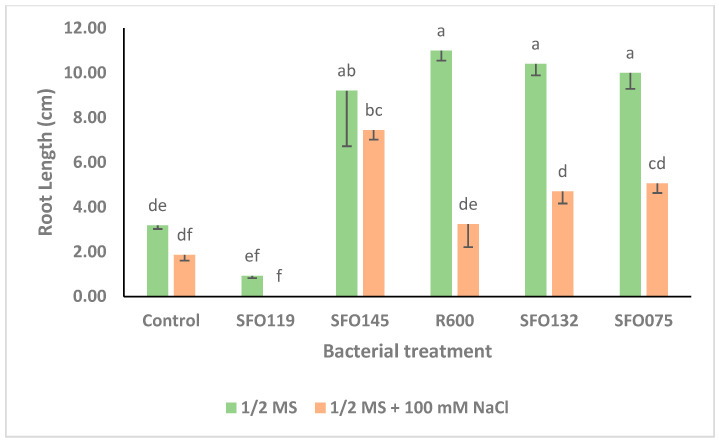
The effect of selected PGPR inoculants on the root length (cm) of maize seeds under saline conditions (1/2 MS + 100 mM NaCl) and non-saline (1/2 MS) conditions. The error bars indicate the root length (cm) mean ± SE. The lettering indicates the significance values among the means by the LSD test (*p* ≤ 0.5).

**Figure 5 plants-14-01107-f005:**
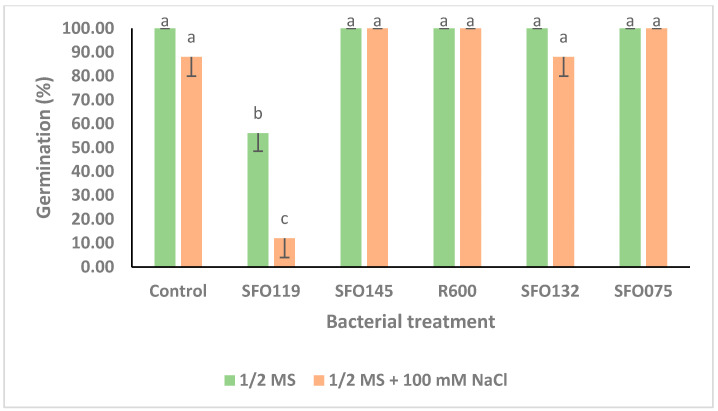
Effect of selected PGPR isolates on germination (%) of maize seeds under saline conditions (1/2 MS + 100 mM NaCl) and non-saline (1/2 MS) conditions. Error bars indicate germination (%) mean ± SE. Lettering indicates significance values among means by LSD test (*p* ≤ 0.5).

**Figure 6 plants-14-01107-f006:**
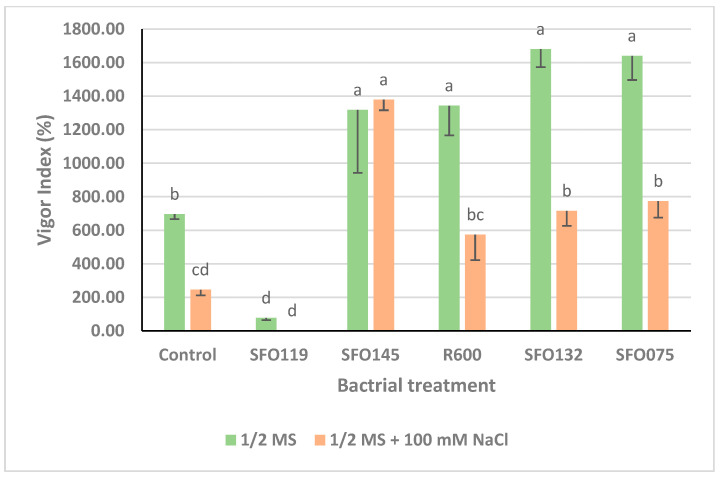
Effect of selected PGPR inoculants on vigor index (%) of maize seeds under saline conditions (1/2 MS + 100 mM NaCl) and non-saline (1/2 MS) conditions. Error bars indicate vigor index (%) mean ± SE. Lettering indicates significance status among means by LSD test (*p* ≤ 0.5).

**Figure 7 plants-14-01107-f007:**
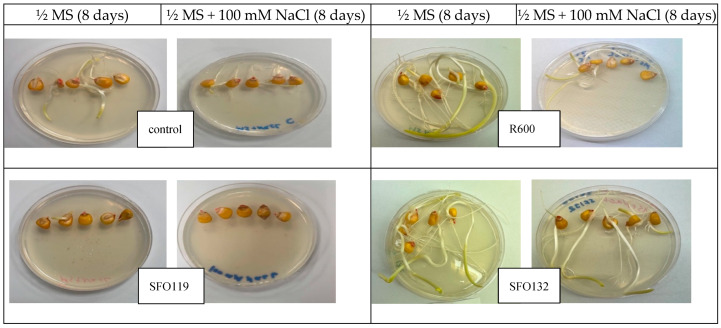
Images of surface-disinfected maize seeds inoculated with five PGPR bacterial strains: SFO119, SFO145, R600, SFO132, and SFO075. Left: non-salt (½ MS, 8 days) and right: salt (½ MS + 100 mM NaCl, 8 days) conditions.

**Table 1 plants-14-01107-t001:** Plant growth-promoting (PGP) traits of selected bacterial strains.

S.No	Isolates	Identification Based on 16S rRNA Sequencing	PGP Traits					NaCl Tolerance				Temperature		
			ACCD	Nfix	P-Sol	IAA	NH_3_	0%	2.5%	5%	10%	37 °C	45 °C	50 °C
1.	SFO075	*Salinicola halophilus*	+	++	+	+	+	+++	+++	+++	+++	+++	−	−
2.	Z180	*Bacillus spizienii*	++	+++	++	+++	+	++	++	++	+	+++	+++	+++
3.	SFO145	*Bacillus haynesii*	+	+++	+	+	++	+++	+++	+++	+++	+++	+++	+++
4.	K2	*Bacillus stercoris*	++	+++	++	+	+	++	+++	++	−	+++	+++	+
5.	K26	*Pseudomonas thivervalensis*	++	++	+	++	+++	++	+++	++	−	+++	+++	++
6.	SFO091	*Staphylococcus pasteuri*	+++	+++	−	+++	+++	+++	+	−	−	+++	−	−
7.	SFO098	*Staphylococcus pasteuri*	++	+	−	+++	++	++	+++	++	−	+++	−	−
8.	K21	*Staphylococcus pasteuri*	+	++	−	++	−	++	+++	++	−	+++	−	−
9.	SFO16	*Psychrobacter faecalis*	−	−	−	+	+++	++	+++	+++	+	+++	−	−
10.	SFO043	*Cytobacillus oceanisediminis*	−	−	−	+	+++	+++	+++	+++	++	+++	+++	++
11.	K13	*Staphylococcus pasteuri*	++	+++	+	+	++	++	++	+	−	+++	−	++
12.	SFO021	*Bacillus subtilis*	+++	+++	+++	−	+++	+++	+++	+++	+++	+++	+++	+++
13.	SFO121	*Metabacillus indicus*	++	+++	+++	−	+++	+++	+++	+++	+++	+++	+++	+++
14.	K20	*Cellulomonas pakistanensis*	++	+++	++	+	−	++	+++	++	−	+++	−	−
15.	SFO037	*Bacillus paramycoides*	+	+++	−	+	+	++	+++	+++	+++	+++	−	−
16.	SFO059	*Ensifer adhaerens*	−	+++	−	++	++	+++	++	+	+	+++	+	+
17.	SFO085	*Nocardioides luteus*	++	+++	−	++	++	+++	+++	+++	++	+++	−	−
18.	SFO106	*Bacillus velezensis*	+++	+++	−	+	+++	+++	+++	++	+	+++	++	+
19.	SFO112	*Staphylococcus pasteuri*	+++	+++	++	−	++	+++	+++	+++	+++	+++	+++	+++
20.	SFO119	*Bacillus inaquosorum*	+++	+++	+	−	++	+++	+++	+++	++	+++	+++	+
21.	SFO133	*Staphylococcus pasteuri*	+++	+++	+	−	+++	+++	+++	+++	+++	+++	+++	+++
22.	B1105	*Arthrobacter cheniae*	++	++	−	−	−	++	+++	+++	+++	+++	+++	++
23.	K22	*Bacillus paralicheniformis*	+	++	−	++	+++	+++	+++	+++	+++	+++	+++	+++
24.	SFO063	*Bacillus haikouensis*	−	−	−	+	−	+++	+++	+++	+++	+++	−	−
25.	SFO074	*Kocuria palustris*	+	+	−	+	++	+++	+++	+++	++	+++	−	−
26.	SFO132	*Staphylococcus petrasii*	++	−	+	+	++	+++	+++	+++	+++	+++	−	−
27.	SFO134	*Staphylococcus pasteuri*	+++	++	++	−	+++	+++	+++	+++	+++	+++	++	++
28.	R600	*Pseudomonas soyae*	−	+++	−	+	+	+++	++	++	−	+++	−	−
29.	SFO041	*Arthrobacter crystallopoietes*	−	−	−	+	++	+++	+++	++	−	+++	−	−
30.	SFO033	*Staphylococcus paralicheniformis*	++	++	−	−	+	++	+++	+++	−	+++	+	+

Where; ‘−’ = No activity, ‘+’ = low activity, ‘++’ = medium activity, and ‘+++’ = high activity.

## Data Availability

All sequences obtained in this study were deposited at GenBank under the accession numbers (PQ454066-PQ454079 and PQ461934-PQ461980).

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
