# Peer review of "Native Plant Growth-Promoting Rhizobacteria Containing ACC Deaminase Promote Plant Growth and Alleviate Salinity and Heat Stress in Maize (*Zea mays* L.) Plants in Saudi Arabia"

_plants, 2025, doi:10.3390/plants14071107_

Round 1

Reviewer 1 Report

Comments and Suggestions for Authors

Dear Authors

This study is very interesting and the authors made a lot of work, a lot of measurements and the had good strategy. Of course all these were in vitro experiments. In a future publication, greenhouse and field experiments should be performed to evaluate the effectiveness of the bacteria in more realistic conditions and determine their impact on crop yield.

I have some comments:

  • Abstract: too long, a lot of details with no need. Summarize it and focus at the main results
  • 2.3. Indole 3-acetic acid (IAA) production: you write: “… To quantify the IAA, absorbance measurements were taken at 535 nm using a UV/Visible spectrophotometer. The IAA concentration was estimated with a standard curve of IAA.”

The IAA activity of the bacterial strains was assessed according to the instructions of Ribeiro and Cardoso. The formation of a pink color indicated the production of IAA.

But If you are indeed measuring the quantity of IAA (UV/Visible spectrophotometer), where are the measurements per strain or even the graph?

  • There is no analysis of soil composition. The study generally states (from the references) that Saudi Arabian soils are sandy with low organic matter and high salt content. However, no detailed information is provided on the composition of the soil from which the bacteria were isolated. This could affect the results, as the soil composition may influence the growth and activity of the bacteria. If you have info mention it
  • Although Tables 2 and 3 provide excellent information, they are a bit chaotic and could be included in the supplementary or stacked bar chart  or heatmap

Author Response

Responses to reviewer’s comments

Reviwer#1

This study is very interesting and the authors made a lot of work, a lot of measurements and the had good strategy. Of course, all these were in vitro experiments. In a future publication, greenhouse and field experiments should be performed to evaluate the effectiveness of the bacteria in more realistic conditions and determine their impact on crop yield.

We sincerely appreciate the reviewer’s # 1 positive feedback on our manuscript and his recognition of the extensive experimental data presented. We value your insightful comments and suggestions, which have helped us refine our manuscript to ensure a more detailed and accurate interpretation of the results. In fact, most promising isolates in this study are currently evaluated under greenhouse conditions to test its efficiency.

Below, we have addressed each of your concerns carefully to enhance the clarity and scientific rigor of the manuscript.

 Comments:

Abstract: too long, a lot of details with no need. Summarize it and focus at the main results

Response: Done.

2.3. Indole 3-acetic acid (IAA) production: you write: “… To quantify the IAA, absorbance measurements were taken at 535 nm using a UV/Visible spectrophotometer. The IAA concentration was estimated with a standard curve of IAA.” The IAA activity of the bacterial strains was assessed according to the instructions of Ribeiro and  Cardoso. The formation of a pink color indicated the production of IAA. But If you are indeed measuring the quantity of IAA (UV/Visible spectrophotometer), where are the measurements per strain or even the graph?

Response: We quantified IAA for only selected isolates in the study. Thus, we did not include the results data of quantification of IAA in the manuscript. The reason behind we quantified IAA for only selected isolates is that we are conducting another experiment using the most promising isolates in this study under greenhouse condition. Therefore, we kept the data of quantification of IAA for the upcoming publication. For the sake of the reviewer, we include the results of quantification of IAA in the supplementary files section (Table S4).

There is no analysis of soil composition. The study generally states (from the references) that Saudi Arabian soils are sandy with low organic matter and high salt content. However, no detailed information is provided on the composition of the soil from which the bacteria were isolated. This could affect the results, as the soil composition may influence the growth and activity of the bacteria. If you have info mention it

Response: The composition of soils from which bacteria were isolated have been provided into supplementary (Table S3).

Although Tables 2 and 3 provide excellent information, they are a bit chaotic and could be included in the supplementary or stacked bar chart or heatmap

Response: Done: We modified Table 3 to include all information presented in Table1 &2. Then we removed Tables 1 &2 to supplementary files section. Also, many figures were removed to supplementary files section.

Reviewer 2 Report

Comments and Suggestions for Authors

This manuscript is focused on the PGPR screening and characterization and their potentiality assessment under abiotic stress conditions. The concept is encouraging; however, the presentation seems to be bulky. The following points need to be addressed before further proceedings:

  1. The title should be revised. Include "native PGPR" and place "Saudi Arabia" at the end of the title, or replace it suitably.
  2. Please be concise in the abstract. Squeeze the methodology and key findings and make it more lucrative, avoiding the bulkiness.
  3. Rearrange the introduction section. The author may follow the chronology (Maize, PGPR, application of PGPR to alleviate stress conditions). Please add a strong hypothesis of the current bottleneck of PGPR studies in the real field under the diverse field conditions.
  4. Results section is too bulky with numerous figures. Thus, it is difficult to follow the trend and novelty of this study. Please avoid well-explored findings in the main text, keeping only the vital ones. The author may shift the associated results and figures to a supplementary file.
  5.  Add the current research pitfall in the discussion section.
  6. Revise the conclusion section. Please add key findings and specific recommendations, such as PGPR consortium, effective microbiomes (EM), etc.
  7. The plagiarism shows that 30% similarity. I recommend reducing it by 20%. However, the plagiarism issue is finally decided by the editorial office of the journal.

Author Response

Reviwer#2

This manuscript is focused on the PGPR screening and characterization and their potentiality assessment under abiotic stress conditions. The concept is encouraging; however, the presentation seems to be bulky. The following points need to be addressed before further proceedings:

Response: We sincerely appreciate your positive evaluation of our work and your encouraging comments regarding the quality and significance of our manuscript.

We have carefully addressed all your comments and suggestions, making the necessary revisions to improve the manuscript accordingly. Below, we provide detailed responses to each point you have raised, along with the corresponding modifications in the text.

Thank you again for your time and constructive feedback, which has helped us enhance the clarity and scientific rigor of our study.

Comments:

The title should be revised. Include "native PGPR" and place "Saudi Arabia" at the end of the title, or replace it suitably.

Response: Done. The title has been revised.

Please be concise in the abstract. Squeeze the methodology and key findings and make it more lucrative, avoiding the bulkiness.

Response: Done.

Rearrange the introduction section. The author may follow the chronology (Maize, PGPR, application of PGPR to alleviate stress conditions). Please add a strong hypothesis of the current bottleneck of PGPR studies in the real field under the diverse field conditions.

Response: Done. The introduction section has been rearranged completely. In addition of bottleneck of PGPR studies in the real field.

Results section is too bulky with numerous figures. Thus, it is difficult to follow the trend and novelty of this study. Please avoid well-explored findings in the main text, keeping only the vital ones. The author may shift the associated results and figures to a supplementary file.

Response: Done. Results section was modified and many tables and figures were removed to the supplementary files.

Add the current research pitfall in the discussion section

Response: Done. We have added paragraph in the Discussion section to address the current pitfalls.

Revise the conclusion section. Please add key findings and specific recommendations, such as PGPR consortium, effective microbiomes (EM), etc.

Response: Done.

The plagiarism shows that 30% similarity. I recommend reducing it by 20%. However, the plagiarism issue is finally decided by the editorial office of the journal.

Response: We rewrite some sections in the manuscript and rearrange the introduction section completely. We hope this reduce the similarity level. Indeed, if the manuscript gets accepted for publication, we can reduce the similarity level further more if necessary, upon the request of the editor.

Reviewer 3 Report

Comments and Suggestions for Authors

The manuscript entitled “Plant Growth-Promoting Rhizobacteria containing ACC Deaminase in Saudi Arabia Promote Plant Growth and Alleviate Salinity and Heat Stress in Maize (Zea mays L.) Plants” aims to characterized 66 plant growth-promoting bacterial isolates, originating from various habitats in Saudi Arabia, for their effect on  germination and growth of maize seeds under normal and three different NaCl concentrations (2.5%, 5%, and 10%) and three different temperatures (37, 45, and 50 °C). The authors identified PGPR isolates with tolerance to 10% NaCl and high temperatures. In in vitro experiments of germination under salinity and non-salinity conditions, the four bacrterial strains Bacillus haynesii (SFO145), Salinicola halophilus (SFO075), Staphylococcus pasteuri (SFO132), and Pseudomonas soyae (R600) exhibited a significant increase in the vigor index of maize plants compared to the un-inoculated control.

General comments:

In my opinion the manuscript is clear and well written. The introduction provides sufficient background, and cites key publications. The work is comprehensible to scientists outside the topic of the work. The research design seems appropriate and conclusions are supported by the results. I have some suggestion to improve and complete the paper

Specific comments:

In introduction

Lines 90-91: I suggest adding data to support and complete this sentence about the maize importance

In Materials and Methods

Line 456: there is no correspondence between cited reference (Nautiyal) and reference number 84 (Kumari et al., 2018). Please, correct it.

Line 475: Please, specify the spectrophotometer used and the IAA quantities used to make standard curve.

Line 497: Please, delete “)” after 1.5 g.

Lines 507-508: Please, add details about the number and types of maize varieties used, their origin, the number of seeds tested, the number of repetitions ecc…

Line 545-546: I suggest writing only one of the two formulas on the “Vigor index”

Author Response

Reviwer#3

The manuscript entitled “Plant Growth-Promoting Rhizobacteria containing ACC Deaminase in Saudi Arabia Promote Plant Growth and Alleviate Salinity and Heat Stress in Maize (Zea mays L.) Plants” aims to characterized 66 plant growth-promoting bacterial isolates, originating from various habitats in Saudi Arabia, for their effect on  germination and growth of maize seeds under normal and three different NaCl concentrations (2.5%, 5%, and 10%) and three different temperatures (37, 45, and 50 °C). The authors identified PGPR isolates with tolerance to 10% NaCl and high temperatures. In in vitro experiments of germination under salinity and non-salinity conditions, the four bacrterial strains Bacillus haynesii (SFO145), Salinicola halophilus (SFO075), Staphylococcus pasteuri (SFO132), and Pseudomonas soyae (R600) exhibited a significant increase in the vigor index of maize plants compared to the un-inoculated control.

General comments:

In my opinion the manuscript is clear and well written. The introduction provides sufficient background, and cites key publications. The work is comprehensible to scientists outside the topic of the work. The research design seems appropriate and conclusions are supported by the results. I have some suggestion to improve and complete the paper

Response: We sincerely appreciate your positive evaluation of our work and your encouraging comments regarding the quality and significance of our manuscript.

We have carefully addressed all your comments and suggestions, making the necessary revisions to improve the manuscript accordingly. Below, we provide detailed responses to each point you have raised, along with the corresponding modifications in the text.

Specific comments:

In introduction

Comment: Lines 90-91: I suggest adding data to support and complete this sentence about the maize importance

Response: Done. More data were added to the manuscript.

In Materials and Methods

Comment: Line 456: there is no correspondence between cited reference (Nautiyal) and reference number 84 (Kumari et al., 2018). Please, correct it.

Response: Done. Reference was corrected.

Comment: Line 475: Please, specify the spectrophotometer used and the IAA quantities used to make standard curve.

Response: Done. Required information were added to the manuscript.

Comment: Line 497: Please, delete “)” after 1.5 g.

Response: Done.

Comment: Lines 507-508: Please, add details about the number and types of maize varieties used, their origin, the number of seeds tested, the number of repetitions ecc…

Response: Done. Required information were added to the manuscript.

Comment: Line 545-546: I suggest writing only one of the two formulas on the “Vigor index”

Response: Done. One Formula was kept in the manuscript.

Round 2

Reviewer 2 Report

Comments and Suggestions for Authors

The revised manuscript have addressed all the concerns raised by the reviewer. This revised manuscript is now suitable for publication.